# Regulation of Three Key Kinases of Brassinosteroid Signaling Pathway

**DOI:** 10.3390/ijms21124340

**Published:** 2020-06-18

**Authors:** Juan Mao, Jianming Li

**Affiliations:** 1State Key Laboratory for Conservation and Utilization of Subtropical Agro-Bioresources, South China Agriculture University, Guangzhou 510642, China; maojuan@scau.edu.cn; 2Guangdong Key Laboratory for Innovative Development and Utilization of Forest Plant Germplasm, College of Forestry and Landscape Architecture, South China Agricultural University, Guangzhou 510642, China; 3Department of Molecular, Cellular, and Developmental Biology, University of Michigan, Ann Arbor, MI 48109, USA

**Keywords:** brassinosteroids, receptor-like kinases, GSK3-like kinases, somatic embryogenesis receptor-like kinases, protein phosphatases

## Abstract

Brassinosteroids (BRs) are important plant growth hormones that regulate a wide range of plant growth and developmental processes. The BR signals are perceived by two cell surface-localized receptor kinases, Brassinosteroid-Insensitive1 (BRI1) and BRI1-Associated receptor Kinase (BAK1), and reach the nucleus through two master transcription factors, bri1-EMS suppressor1 (BES1) and Brassinazole-resistant1 (BZR1). The intracellular transmission of the BR signals from BRI1/BAK1 to BES1/BZR1 is inhibited by a constitutively active kinase Brassinosteroid-Insensitive2 (BIN2) that phosphorylates and negatively regulates BES1/BZR1. Since their initial discoveries, further studies have revealed a plethora of biochemical and cellular mechanisms that regulate their protein abundance, subcellular localizations, and signaling activities. In this review, we provide a critical analysis of the current literature concerning activation, inactivation, and other regulatory mechanisms of three key kinases of the BR signaling cascade, BRI1, BAK1, and BIN2, and discuss some unresolved controversies and outstanding questions that require further investigation.

## 1. Introduction

Brassinosteroids (BRs) are plant-specific steroid hormones and play essential roles in a broad range of plant growth and developmental processes, including cell elongation, cell division, and differentiation, seed germination, stomata formation, root development, vascular differentiation, plant architecture, flowering, male fertility, and senescence [1,2,3,4]. BRs are also involved in responding to various abiotic and biotic stresses, such as drought, flooding, salinity, extreme temperatures, microbial pathogens, and insect herbivores [5,6,7]. Plants with defects in BR biosynthesis or signaling show a characteristic set of developmental defects, including dwarfed statue, male sterility, delayed senescence and flowering, and photomorphogenesis in the dark [8,9].

Using various experimental and theoretical approaches, including genetics, biochemistry, cell biology, chemical biology, structural biology, proteomics, transcriptomics, genomics, mathematical modeling, and computational dynamics simulation, a series of important BR signaling components have been well established and intensively studied, revealing a protein phosphorylation-mediated BR signaling cascade [10,11,12] (Figure 1). BRs are perceived at the plasma membrane (PM) by the extracellular domains of BRI1 (Brassinosteroid-Insensitive1) receptor, a leucine-rich repeat receptor-like kinase (LRR-RLK) [9,13], and its co-receptor BAK1 (BRI1-Associated receptor Kinase1, also known as SERK3 for Somatic Embryogenesis Receptor Kinase3) [14,15,16], a versatile LRR-RLK involved in many signaling processes [17]. BR binding to the BR-binding pocket formed by the extracellular domains of BRI1 and BAK1 is thought to trigger conformational changes of their cytoplasmic domains [18,19]. BRI1 subsequently phosphorylates its inhibitor BKI1 (BRI1 Kinase Inhibitor1) and induces its dissociation from the PM [14,18,20,21], thus enabling heterodimerization, reciprocal phosphorylation, and full activation of the kinase activities of BRI1 and BAK1 [14,18,20,21,22,23]. The fully activated BRI1 triggers a series of phosphorylation or dephosphorylation events to transduce the extracellular BR signals into the cytosol. BRI1 phosphorylates BSK1 (BR-Signaling Kinase1), CDG1 (Constitutive Differential Growth1), and some of their homologs, leading to phosphorylation and subsequent activation of members of a unique family of protein phosphatases with Kelch-like domain (PPKLs) that include BSU1 (bri1 suppressor1) and BSU1-Like1-3 (BSL1-3) [24,25,26]. It is generally believed that the phosphorylated BSU1/BSLs inactivate BIN2 (Brassinosteroid-Insensitive2), which is a member of the plant GSK3 (Glycogen Synthase Kinase3)-like kinase family, via dephosphorylation of a phosphorylated tyrosine (Tyr) residue in the activation loop of BIN2 [27]. The dephosphorylated BIN2 also interacts with KIB1 (Kink suppressed in *bzr1-1D*1), an F-box E3 ubiquitin ligase, leading to BIN2 ubiquitination and proteasome-mediated degradation [28]. Upon BIN2 inactivation and degradation, two highly similar BIN2 substrates, BZR1 (Brassinazole-resistant1) and BES1 (bri1-EMS suppressor1) [29,30] are rapidly dephosphorylated by certain nuclear-localized members of the PP2A (protein phosphatase 2A) family [31], leading to their nuclear accumulation. The dephosphorylated BZR1 and BES1 bind to their target promoters containing BRRE (BR-response element) (CGTGC/TG) and/or E-box (CANNTG) motif to regulate expression of thousands of BR-responsive genes that are crucial for plant growth and development [32,33].

In the past few years, several excellent reviews were published that summarized the significant progresses in understanding how the BR signal is perceived at the cell surface and how the extracellular BR signal is transduced into the nucleus to regulate a wide range of plant developmental and physiological processes [10,11,12,34]. In this review, we present a critical analysis of currently available data on three early BR signaling components, including the BR receptor BRI1, its coreceptor BAK1, and the crucial negative regulator BIN2, highlighting recent findings and discussing controversies and unanswered questions on the regulatory mechanisms that control their protein abundance, subcellular localizations, and signaling activities.

## 2. BRI1, the BR Receptor

BRI1 localizes to the PM and belongs to a large and plant-specific family of LRR-RLKs, which is composed of 223 members in *Arabidopsis thaliana* [35]. BRI1 consists of an extracellular LRR domain of 25 LRRs, a single-pass transmembrane segment, and a cytoplasmic kinase domain [9]. The 25 LRRs are interrupted by a 70-residue island domain (ID, from amino acid Lys^586^ to Met^657^), which constitutes the BR binding domain with the 22nd LRR [36]. Its cytoplasmic kinase domain can be subdivided into the JM (juxtamembrane) region, a canonical serine/threonine (Ser/Thr) kinase domain, and a short C-terminal tail (CT) of 36 amino acids (AAs). The Arabidopsis genome encodes three BRI1 homologs: BRL1 (BRI1-Like1), BRL2 (also known as VH1 for Vascular Highway1 [37]), and BRL3. BRL1 and BRL3 can also bind BRs and function as BR receptors [38,39], whereas BRL2/VH1 does not bind BR but functions in a BR-independent manner to regulate vascular development [37,40]. Gene expression analysis and genetic studies revealed that BRI1 is expressed in most plant tissues/organs, whereas BRLs are found only in the vascular tissues and stem cell niches [41]. BRLs have been shown to be involved in vascular development [42] and plant tolerance to abiotic stresses, such as hypoxia and drought [43,44].

### 2.1. Maintaining the Inactive State in the Absence of BR

In the absence of BR ligands, BRI1 is maintained at its inactive state via its inhibitory CT and its binding to BKI1 at the PM [20,23]. Removal of the BRI1 CT not only increased the in vitro kinase activity of a recombinant BRI1 kinase protein, but also led to a hyperactive BR receptor in vivo [23]. Further investigation will likely be required to fully understand the mechanism of this autoinhibitory activity of the BRI1 kinase activity given the missing CT in several available crystal structures of the BRI1 kinase domain [45,46]. The 36-AA CT could block the substrate binding site to interfere with the in vitro homodimerization for the auto(trans)phosphorylation activity or the in vivo binding of the kinase domains of BRI1 and BAK1. In addition to the “*cis*” autoinhibition, the binding of the PM-associated BKI1 to the BRI1 cytoplasmic domain demonstrates a “*trans*” inhibitory mechanism to prevent association and subsequent cross phosphorylation of the kinase domains of BRI1 and BAK1, which is likely triggered by BR-independent heterodimerization of their extracellular domains known to occur in plant cells [15,47,48]. A previous genetic demonstration of the absolute requirement of BAK1/SERKs for the BRI1’s activation [49] implies that BRI1 can be kept in its inactive state via competitive binding of BAK1 with certain BAK1-interacting proteins. Indeed, recent studies revealed that ligand-independent interaction of BAK1 with BIR3 (BAK1-Interacting Receptor-like kinase3) could inhibit BR-stimulated BAK1-BRI1 heterodimerization and BRI1 activation (see below for more discussion on the BAK1-BIR interaction) [50].

### 2.2. BRI1 Activation

It was well known that many kinases are activated via phosphorylation of crucial Ser/Thr residues in the activation segment catalyzed by upstream kinases [51]. In animal cells, many receptor kinases are activated by homodimerization and auto(trans)phosphorylation [52,53]. Despite several reports of BRI1 homodimerization in vitro or in vivo [15,23,54,55], BRI1 activation requires its heterodimerization and subsequent transphosphorylation with its coreceptor BAK1/SERKs. An earlier in vitro kinase assay with dephosphorylated recombinant kinases of BRI1 and BAK1 showed that neither BRI1 nor BAK1 was active when incubated alone with the ATP-containing kinase reaction buffer but became active when incubated together [45]. Similarly, the full-length BRI1 or BAK1 was inactive when expressed alone in yeast cells but became active kinases when they were coexpressed. Neither BRI1 nor BAK1 was active when coexpressed in yeast with a kinase-dead partner [18]. More important, a BRI1-FLAG fusion protein could not be activated by exogenously applied BR in a transgenic Arabidopsis line that lacks BAK1 and its two close homologs [49]. Furthermore, a recent study showed that ligand-independent heterodimerization of a chimeric LRR-RLK, which was composed of the extracellular domain of BIR3 and the intracellular domain of BRI1, and BAK1 led to constitutive activation of the BR signaling pathway [56]. All these biochemical and genetic/transgenic experiments argue against the widely accepted sequential-phosphorylation model. This model posits that BR binding-triggered conformational changes result in weak activation of BRI1 and its subsequent phosphorylation and activation of BAK1, which can then transphosphorylate BRI1 for its full activation [21]. Further studies, especially structural analysis of the ligand-bound full-length BRI1-BAK1 complexes and mass spectrometry (MS)-based quantitative phosphorylation analysis of BRI1 in *bak1/serk* mutants, are needed to fully understand the activation mechanism of BRI1 and the genetic requirement of BAK1/SERKs for its activation.

Structural analyses of the BRI1-BAK1 extracellular heterodimers and molecular dynamics simulations suggested that BR binding to the extracellular domain of BRI1 causes conformational changes in BRI1 [57,58]. Such subtle structural changes not only stabilize the ID to create a docking platform for the BRI1-BAK1 association, which is further stabilized by a bound active BR (functioning as “molecular glue” to interact with the N-terminal cap of BAK1), but also potentially create a secondary interface involving the N-terminal 12 LRRs of BRI1 to interact with BAK1 [16,19] (Moffett and Shukla, 2020 bioRxiv: http://doi.org/10.1101/630640). Despite lack of experimental evidence or structural information, it has been widely accepted that stable association of the extracellular domains of BRI1 and BAK1 brings their cytoplasmic domains into close proximity, thus permitting cross phosphorylation, especially at Ser^1044^ of BRI1 and Thr^450^ of BAK1 within their respective activation loop, and subsequent activation of both kinases [45,46,59]. A recent structural modeling study suggested that the cytoplasmic domains of BRI1 and BAK1 could weakly interact independently of BR binding to their extracellular domains [60]. However, it is important to point out that the structural models of the kinase domains of BRI1 and BAK1 used for the modeling study were derived from autophosphorylated and activated kinase domains with stabilized αC-helixes and activation segments.

Structural models of the activated BRI1 kinase domain [45,46] suggested that the phosphorylated Ser^1044^ is likely trapped in a positively-charged phosphate-binding pocket consisting of Arg^922^ (at the beginning of the αC-helix), Arg^1008^ [within the HRD (His-Arg-Asp) motif], and Arg^1032^ (at the beginning of the β9-strand), thus stabilizing the activation loop and establishing an active kinase conformation (Figure 2). Specifically, the interaction of the phosphorylated Ser^1044^ (pSer^1044^) and Arg^922^ of the αC-helix is likely critical for creating the so-called regulatory-spine (R-spine) consisting of 5 residues [the anchoring Asp^1068^ residue of the αF-helix, His^1007^ of the HRD motif, Phe^1028^ of the DFG (Asp-Phe-Gly) motif, Ile^931^ of the αC-helix, and Leu^942^ of the β4-strand] (Figure 2). The assembly of the R-spine has been considered as a crucial indicator of an active conformation of eukaryotic protein kinases [61,62]. As expected, a Ser^1044^-Ala mutation resulted in a strong loss of kinase and signaling activity of BRI1 in vitro and in vivo [22].

In addition to the pSer^1044^ residue within the activation loop, previous studies revealed an essential role of a conserved Ser/Thr residue (Thr^1049^) in the “P+1 loop” [21,22], which is thought to be the docking site for the backbone of the substrate P-site and the side-chain of the P+1 residue [61]. Previous bioinformatic analysis indicated that this position is occupied by a Ser/Thr residue in >99% of the so-called “RD”-type RLKs containing Arg-Asp residues within the conserved HRD motif [21]. By contrast, only 30% of “non-RD”-type RLKs contain a Ser/Thr residue at this position. Mutating Thr^1049^ to Ala completely inhibited the in vitro phosphorylation activity of a recombinant BRI1 kinase domain and greatly reduced its signaling activity in transgenic Arabidopsis plants [23], demonstrating its crucial role in BRI1 kinase activity. Mutating the equivalent Ser/Thr residue in several other “RD”-type LRR-RLKs resulted in loss of in vitro phosphorylation activity [63]. Further studies will be needed to determine whether the phosphorylation of this residue is truly required for BRI1 activation as the phosphorylation-mimic Thr^1049^-Asp mutation also led to a strong loss of the in vitro phosphorylation activity of a recombinant BRI1 kinase [21]. Based on the structural model (Figure 2), the phosphorylation of this residue could potentially interfere with ATP binding and/or substrate binding.

The activated BRI1 is thought to autophosphorylate additional Ser, Thr, and Tyr residues in both N-lobe and C-lobe (Figure 2). The phosphorylated N-lobe residues include Ser^838^, Thr^842^, Thr^846^, Thr^851^, and Ser^858^ of the JM (not shown in Figure 2), Thr^872^ and Thr^880^ on the αB-helix that caps the N-lobe, Ser^887^ and Ser^891^ on both ends of the β1-strand, Ser^906^ in the β2-β3 loop, Ser^917^ in the β3-αC loop (missing in the BRI1 crystal structure), and Thr^930^ on the αC-helix. The phosphorylated C-lobe residues include Ser^963^ at the start of the αD-helix, Ser^981^, Thr^982^, and Ser^990^ of the αE-helix, Ser^1012^ and Ser^1013^ of a half helical twist before the β7-strand, Ser^1026^ located between the β8-strand and the DFG motif, Thr^1039^, Ser^1042^, and Thr^1045^ of the activation segment, Ser^1071^ and Thr^1081^ of the αF-helix, Ser^1109^ located at the start of the twisted helical αG-αH linker, Thr^1147^ at the start of the αI-helix, and at least 5 residues in the CT (Ser^1166^, Ser^1168^, Thr^1169^, Ser^1179^, and Ser^1187^) [21,22,63,64,65,66] (Figure 2). Among those phosphorylated residues, at least 11 were identified from immunoprecipitated BRI1 fusion protein from transgenic Arabidopsis plants, including 6 uniquely identified (Ser^838^, Ser^858^, Thr^872^, Thr^880^, Thr^982^, and Ser^1168^), 2 additional Thr sites (most likely Thr^842^ and Thr^846^) and 3 ambiguous sites within the activation fragment [22]. In addition, at least 3 Tyr residues were reported to be autophosphorylated, including Tyr^831^ (in the JM), Tyr^956^ (the gatekeeper residue on the β5-strand), and Tyr^1072^ (on the αF-helix) [67,68] (Figure 2). It should be noted that the published MS data from several in vitro and in vivo phosphorylation site-mapping experiments of BRI1 did not identify any phosphorylated Tyr residue [21,22,63,64,65,66]. The 3 phosphorylated Tyr residues (Tyr^831^, Tyr^956^, and Tyr^1072^) that were discussed in the current literature were deduced from immunoblot assays with generic and site-specific anti-phosphorylated Tyr (anti-pTyr) antibodies [67,68]. It is generally believed that these phosphorylated Ser/Thr/Tyr sites could regulate the abundance and/or signaling activity of BRI1 and create docking sites for binding downstream signaling components or regulators to build a BRI1 signaling complex, which likely include scaffolding proteins such as TTLs (Tetratricopeptide Thioredoxin-Like proteins) and BSK3 [69,70]. The activated BRI1 can then transphosphorylate these regulators and downstream targets to transduce the extracellular BR signal into the cytosol, ultimately reaching the nucleus to alter gene expression.

One question that remains to be answered is how binding of BR to BRI1 triggers phosphorylation of BKI1. As discussed above, the activation of BRI1’s kinase activity requires its heterodimerization and transphosphorylation with BAK1, a process that is prevented by the PM-localized BRI1-binding BKI1, while BKI1 dissociation from BRI1 and the PM absolutely requires phosphorylation by activated BRI1 [71,72]. Previous studies showed that BRI1-catalyzed Tyr phosphorylation at the conserved Tyr^211^, which likely requires BRI1-catalyzed phosphorylation at Ser^270^ and Ser^274^ [72], is necessary and sufficient for its dissociation from the PM [71]. It might be possible that BR binding to the BRI1’s extracellular domain triggers yet undefined conformational changes in its cytoplasmic domain. Such structural changes could weaken the BRI1-BKI1 binding and permit weak BRI1-BAK1 association to allow the BAK1-catalyzed transphosphorylation of Thr^1044^ in the activation loop of BRI1, leading to further conformational changes and full activation of BRI1. Activated BRI1 can subsequently phosphorylate BKI1 at Ser^270/274^ and Tyr^211^, resulting in BKI1 dissociation from BRI1 and the PM and a stronger BRI1-BAK1 binding. A better understanding of the BRI1-BKI1 and BRI1-BAK1 interaction requires further structural analyses of the protein complexes of full-length proteins in the absence or presence of active BRs.

### 2.3. Attenuation and Deactivation

The magnitude and duration of the BR signaling is dynamically regulated by activation and deactivation of the BR receptor. One simple mechanism of receptor inactivation or attenuation is mediated by autophosphorylation. Previous studies suggested that autophosphorylation at certain residues inhibited the in vitro kinase activity of an *E. coli* expressed BRI1 kinase domain and reduced the physiological activity of BRI1 in transgenic Arabidopsis plants [21,23,67,73]. For example, mutating Thr^872^ to Ala, which is located at the start of the αB-helix that caps the N-lobe of 5 antiparallel β-strands (Figure 2), could significantly enhance the in vitro autophosphorylation activity or transphosphorylation activity towards an artificial peptide substrate [22]. It is interesting to note that Thr^872^ and Thr^880^, known to be phosphorylated and located at the start and end of the αB-helix (Figure 2), are highly conserved among 213 Arabidopsis LRR-RLK [22], suggesting that phosphorylation of these residues might be a conserved attenuation mechanism for many plant LRR-RLKs. Similarly, Ser^891^, located in the β1-β2 ATP-binding loop (better known as “glycine-rich loop” or “G-loop”) containing the GXGXXG-motif (X indicating any AA), was previously shown to deactivate BRI1 [73], likely by interfering with ATP binding, as the “G-loop” is known for positioning the three phosphate groups and the adenine ring of ATP [61]. An earlier study also suggested that phosphorylation of two tyrosine residues, Tyr^831^ (located in the JM) and Tyr^956^ (a critical gatekeeper of the ATP binding pocket), might also inhibit the BRI1 kinase and signaling activity [67]. However, the lack of appropriate phosphorylation-mimic AA for a pTyr residue makes it complicated to interpret the published experimental results on the regulatory roles of BRI1’s Tyr phosphorylation.

In addition to the “*cis*” attenuation by autophosphorylation, the signaling activity of BRI1 could be “*trans*” attenuated by dephosphorylation of certain phosphorylated Ser/Thr residues through PP2A [66,74]. A PP2A holoenzyme is composed of three subunits: a catalytic C subunit, a regulatory B subunit, and a scaffolding A subunit. A B subunit from one of the 3 subfamilies in Arabidopsis: B, B’, and B’’, determines the substrate specificity, while an A subunit brings the B and C subunits together to form an active protein phosphatase complex [75]. An earlier study implicated the Arabidopsis SBI1 (suppressor of bri1), a leucine carboxylmethyltransferase, in methylating the C subunits of PP2A to enhance the PM-association of certain PP2A holoenzymes [74]. As a result, PP2A could bind and dephosphorylate activated BRI1, leading to increased BRI1 degradation and attenuated BR signaling. Further support for the involvement of PP2A in attenuating BR signaling came from a recent study showing that at least four cytoplasm-localized PP2A B’ subunits, including B’γ, B’η, B’θ, and B’ζ [75], interact with BRI1 and mediate its dephosphorylation and inactivation [66]. Overexpression of any of these *PP2A-B’* genes in a weak *bri1-5* mutant significantly enhanced its dwarf phenotype and further reduced its BR signaling activity. By contrast, simultaneously eliminating these four *PP2A-B’* genes could enhance BR signaling and partially suppressed the dwarf phenotype of a weak BR-deficient mutant *det2* (*de-etiolation2*) [66]. Quantitative MS assays of autophosphorylated BRI1 kinase domain incubated with immunoprecipitated PP2A complexes identified several autophosphorylated Ser/Thr residues as the potential dephosphorylation sites of PP2A, including a phosphorylated residue (Thr^872^, Thr^880^, or Ser^887^ on the αB-helix and the αB-β1 loop), Ser^917^ (in the β3-αC loop), Ser^981^ (on the αD helix), and four phosphorylated Ser/Thr residues in the CT (Ser^1166^, Ser^1168^, Thr^1169^, or Ser^1172^, and S^1179^/Thr^1180^) [66]. An earlier mutagenesis experiment indicated that individual mutations of these Ser/Thr residues (except Thr^872^) had little impact on the in vitro autophosphorylation activity of the BRI1 kinase but did reduce its transphosphorylation activity towards a peptide substrate [22]. In addition, the dephosphorylated CT was previously shown to exert inhibitory effects on BRI1 signaling activity [23]. Further MS analyses of the endogenous BRI1 in *PP2A B’*-overexpressing transgenic Arabidopsis lines and the Arabidopsis quadruple mutant lacking B’γ, B’η, B’θ, and B’ζ are needed to pinpoint the exact Ser/Thr residues that are dephosphorylated by PP2A. Given the importance of Tyr phosphorylation in BR signaling [67,68,71] and a previous report of Tyr dephosphorylation of an LRR-RLK immunity receptor EFR (Elongation Factor-Tu Receptor) by a bacterial tyrosine phosphatase [76], it will be interesting to investigate if the BRI1 signaling activity could also be regulated by a member of the Arabidopsis PTP/DSPP (protein tyrosine phosphatase/dual specificity protein phosphatase) family [77].

### 2.4. Regulating the Abundance of BRI1 on the PM

#### 2.4.1. Trafficking from the ER to the PM

The BR signaling activity at the PM is also controlled by the amount of the PM-localized BRI1 receptor, which starts its secretory journey in the endoplasmic reticulum (ER) (Figure 3). It was well known that the ER houses several stringent quality control (QC) systems that permit export of only correctly folded proteins into the Golgi apparatus but retain incompletely or incorrectly folded proteins in the ER for chaperone-assisted folding repair or degradation via ER-associated degradation mechanism (ERAD) [78,79]. Two structurally defect but biochemically competent mutant variants of BRI1, bri1-5 with Cys^69^-Tyr mutation and bri1-9 carrying a Ser^662^-Phe mutation, are retained in the ER and degraded by ERAD [80,81,82] (Figure 3). Loss-of-function mutations in the ER quality control (ERQC) or ERAD system resulted in increased amounts of mutant bri1 receptors on the PM, thus partially suppressing the dwarfism phenotype of these *bri1* mutants as the corresponding mutant BRI1 proteins still retain partial signaling activity after being correctly targeted to the PM [83,84,85,86]. A recent study also suggested the presence of a yet to be defined QC system at the PM to remove misfolded/damaged PM-localized proteins, such as bri1-301 (with a Gly^989^-Ile mutation) that escapes from ERQC/ERAD [87,88]. TWD1 (Twisted Dwarf1), a well-studied cochaperone protein that was previously shown to be localized in the ER and at the PM, where it regulates auxin transport [89], also interacts with both BRI1 and BAK1 to enhance BR signaling [90,91]. Given its demonstrated interaction with HSP90 (heat shock protein 90) [92,93], it is tempting to speculate that TWD1 could promote optimal folding of BRI1 and its coreceptor, thus maximizing their BR signaling activities at the PM.

#### 2.4.2. Targeting to Unique PM Nano/Microdomains

Recent fluorescence microscopy studies coupled with live-cell imaging techniques have revealed that plant LRR-RLKs, including BRI1 and BAK1, are not uniformly distributed on the PM but are rather localized to specific PM subcompartments known as lipid rafts, nanodomains, or microdomains with unique lipid and protein compositions [48,94,95]. It was thought that members of the two protein families: Flotillins (Flot) and plant-specific remorins, might function as organization centers to form protein nanoclusters that are important for receptor signaling [55,95]. Interestingly, AtFlot1, one of the two Arabidopsis Flot homologs, was shown to coexist with BRI1 in a PM microdomain to influence BRI1 endocytosis [55] while OsREM4.1 (*Oryza sativa* remorin4.1), a member of the rice remorin family, interacted with OsSERK1, a rice homolog of BAK1, to regulate the heterodimerization of OsSERK1 with a rice homolog of BRI1, OsBRI1 [96]. It remains to be determined how AtFlot1 and OsREM4.1 recruit BRI1/OsBRI1 to unique nano/microdomains on the PM given the presence of >600 RLKs, but only 2 Flots and 16 remorins in Arabidopsis [97,98].

#### 2.4.3. Endocytosis

Like many PM-localized proteins, the wild-type BRI1 is also known to dynamically undergo clathrin-mediated endocytosis (CME) or clathrin-independent endocytosis (CIE) from the PM to the trans-Golgi network or early endosomes (TGN/EE) [15,55,99], where BRI1 can then be recycled back to the PM or sorted into the vacuole for degradation via multiple multimeric ESCRT (endosomal sorting complex required for transport) complexes [100,101,102,103,104]. Thus, BRI1 endocytosis serves to attenuate BR signaling despite an early report that suggested endosomal initiation of BR signaling [99,105]. CME is a highly conserved cellular process that requires coordination of several groups of proteins, including clathrin triskelia consisting of clathrin heavy chains and light chains, small GTPases and their regulators such as ADP-ribosylation factor-guanine nucleotide exchange factors (ARF-GEFs), and adapter complexes, including the canonical heterotetrameric adapter protein 2 (AP2)-complex and the heterooctomeric TPLATE complex (TPC) [106]. Consistent with this, inhibition of BRI1 endocytosis by tyrphostin A23, which is a widely-used chemical that specifically blocks the cargo recruitment step of CME [107], could inhibit BRI1 endocytosis and enhance BR signaling [55,104]. Similarly, genetic mutations or transgenic interferences (dominant-negative and/or gene silencing) of the CME components, including clathrins, Rab GTPases, two ARF-GEFs (GNOM and GNOM-LIKE1), the AP2-complex, and the TPC complex, impaired BRI1 endocytosis, leading to increased BRI1 abundance at the PM and enhanced BR signaling [104,108,109]. It was generally thought that the CME-mediated BRI1 internalization is a constitutive process; however, a recent study suggested that BRI1 internalization could be stimulated by BR treatment, which is mediated by a CIE pathway involving a membrane microdomain-associated protein AtFlot1 (*Arabidopsis thaliana* Flot1 [110]. AtFlot1 directly associates with BRI1 and interference with AtFlot1 protein affects BRI1 endocytosis, leading to increased BRI1 on the PM and enhanced BR signaling [55]. It remains to be determined to what degree coordination of the CME and CIE pathways regulate BRI1 endocytosis and downstream BR signaling.

#### 2.4.4. The Endocytic Pathway to the Vacuole for Degradation

Endocytosed BRI1 proteins accumulate in the TGN/EE compartments where they are sorted into the vacuole for degradation, an endocytic pathway that is mediated through sorting/packaging of BRI1 into ILVs (intraluminal vesicles) of the LE/MVBs (late endosomes/multivesicular bodies) and eventual MVB-vacuole fusion [111]. Previous studies suggested that BRI1 endocytosis and its subsequent sorting at TGN/EE into MVBs is likely regulated by the E3 ligases PUB12/13 (plant U-box protein12/13)-catalyzed ubiquitination of BRI1 [112] while its packaging into MVBs is regulated by ESCRT protein complexes and their associated proteins [113]. Interestingly, BR treatment stimulated the BRI1-PUB13 interaction and PUB13 phosphorylation and simultaneous elimination of PUB12 and PUB13 inhibited BRI1 internalization, leading to increased BRI1 abundance and enhanced BR sensitivity [112]. It was known that K63-linked polyubiquitination is involved in endocytosis and subsequent ESCRT-mediated vacuolar delivery, whereas K48-linked polyubiquitination is the signal for proteasome-mediated degradation [114,115]. A previous study indicated that PUB12/13-catalyzed ubiquitination of FLS2 (Flagellin Sensing2), another LRR-RLK plant immunity receptor that senses bacterial flagellin [116], is involved in proteasome-mediated FLS2 degradation [117], implying that PUB12/13 likely catalyzes the K48-linked polyubiquitination of FLS2. Thus, it will be interesting to determine the exact type of the ubiquitin-linkage and the exact site(s) of the PUB12/13-catalyzed BRI1 ubiquitination given an earlier report indicating that the K63-linked polyubiquitination at Lys^866^ of an immunoprecipitated BRI1 is the likely signal to drive the BRI1 endocytosis [115]. Experiments are also needed to fully understand the discrepancy between the BR-dependent BRI1-PUB12/13 interaction [112] and the kinase-dependent but ligand-independent BRI1 ubiquitination [115].

Interestingly, a partial loss-of-function mutation in Arabidopsis ALIX (apoptosis-linked gene 2-interacting protein X), which is required for localizing an Arabidopsis deubiquitinating enzyme in LE/MVBs and associates with an ESCRT complex to mediate packaging of cargos into ILVs [118], was found to be defective in the vacuolar delivery of BRI1 [113]. It remains unknown whether the failure to package endocytosed BRI1 into ILVs is functionally related to the mislocalization of the deubiquitinating enzyme in the *alix* mutant. A recent study also implicated BIL4 (Brassinazole-Insensitive-Long hypocotyl4), a 7-transmembrane protein localized in the TGN/EE, LE/MVB, and vacuolar membrane, in regulating the endocytic trafficking of BRI1 to the vacuole [119]. RNAi-mediated *BIL4* silencing resulted in increased BRI1 localization in the vacuole, whereas *BIL4* overexpression reduced the BRI1 trafficking from the TGN/EE to the vacuole. It remains to be investigated to fully understand the biochemical function of BIL4 in the plant endocytic pathway.

#### 2.4.5. Endocytic Recycling

Most of the endocytosed BRI1 are thought to be recycled back to the PM to replenish the PM pool of the BR receptor, thus enhancing BR signaling [99,101,102,104]. It was thought that the TGN/EE-PM recycling involves retromer complex-mediated cargo retrieval, microtubule-assisted trafficking of recycling endosomes, and SNARE (soluble N-ethylmaleimide-sensitive factor attachment proteins receptor)-mediated exocytosis [111]. A hypomorphic allele of *DET3* that encodes the cytosolic C subunit of the Arabidopsis vacuolar ATPase (V-ATPase) reduces the ability of V-ATPase to acidify the TGN/EE but not the Golgi or the vacuole, leading to compromised secretion and recycling of BRI1 and reduced BR sensitivity [120,121]. An Arabidopsis microtubule-associated protein CLASP (cytoplasmic linker protein-associated protein) was previously known to interact with SNX1 (sorting nexin1), a component of the Arabidopsis retromer complex, to mediate the endosome-microtubule association [122]. Interestingly, the expression of CLASP is regulated by BR in a BRI1-dependent manner and CLASP also mediates the BR-induced microtubule reorganization [123]. Importantly, a loss-of-function mutation in CLASP compromised the TGN/EE-PM trafficking of the constitutive cycle of BRI1 endocytosis-exocytosis, leading to reduced BRI1 abundance on the PM and dampened BR sensitivity [123]. The exocytosis is known to be coordinated by Rab GTPase, the vesicle tethering complex known as exocyst, and SNAREs [124]. Mutations in EXO70A1 (exocyst subunit 70A1, a component of the Arabidopsis exocyst complex) or components of the SNARE complex reduced BRI1 recycling back to the PM [125,126]. A recent study also implicated BIG3 (brefeldin A-inhibited guanine nucleotide-exchange protein3) and BIG5, two members of the BIG subfamily of the Arabidopsis ARF-GEFs, in BRI1 recycling. Simultaneous elimination of BIG3 and BIG5 resulted in dwarfed plants with reduced BR sensitivity [127], but their exact cellular functions remain to be defined. Given the widely accepted model of constitutive endocytosis of BRI1, it will be interesting to investigate how plant cells integrate developmental cues and environmental signals to balance the endocytic vacuolar degradation and the endocytic recycling process to control the abundance of signaling competent BRI1 on the PM.

## 3. BAK1, the Coreceptor

BAK1/SERK3 and three other members of the SERK family, SERK1, SERK4, and SERK5 (nonfunction in the Col-0 ecotype of *Arabidopsis thaliana* but remains function in the L*er*-0 ecotype [128]), are required to function as the BRI1 coreceptor to initiate the BR signaling at the PM [14,18,129,130]. The 5 members of the SERK family share the same structural organization with an extracellular domain of 5 LRRs plus the N-terminal cap, a single transmembrane helix, a cytoplasmic kinase domain, and a CT with a conserved Ser-Gly-Pro-Arg motif at their C-terminal end known to be important for their kinase activity [131]. Structural analysis of the BRI1-BAK1 heterodimer of their extracellular domains identified key residues involved in forming and stabilizing the BAK1-BRI1 dimer [19]. In addition to functioning as the coreceptors for BRI1 to activate the ligand-bound BRI1, BAK1/SERKs were found to be versatile coreceptors that heterodimerize with many ligand-bound RLKs for their activation [17] and intracellular signal transduction.

### 3.1. Phosphorylation of BAK1

As discussed above, BAK1 activation requires its heterodimerization with BRI1 when assayed with *E. coli* expressed kinase domains of BRI1 and BAK1 [45] or with yeast expressed full-length proteins [18]. An earlier transgenic experiment, which expressed a BAK1-GFP fusion protein in a strong *bri1-1* mutant background, showed that exogenous BR application resulted in no detectable change in BAK1 phosphorylation [21], suggesting that BAK1/SERKs activation in vivo also requires its heterodimerization with BRI1.

MS analyses of *E. coli*-expressed recombinant BAK1 kinase, in vitro phosphorylated forms, and immunoprecipitated BAK1 fusion proteins from transgenic Arabidopsis plants identified a total of 23 phosphorylation sites in both N-lobe and C-lobe (Figure 4). The phosphorylated residues in the N-lobe include Ser^286^ of the αB-helix, Ser^290^ of the αB-β1 loop, Thr^312^ of the β2-β3 loop, Ser^324^ within the β3-αC loop (that is missing in the BAK1 kinase structure), Thr^333^ and Ser^339^ of the αC-helix, and Thr^355^ and Thr^357^ of the β4-β5 loop. The phosphorylated residues of the C-lobe include Ser^370^ and Ser^373^ on the αD-helix, Ser^381^ of the αD-αE loop, Tyr^443^, Thr^446^, Thr^449^, Thr^450^, and Thr^455^ on the activation segment, Ser^465^ and Thr^466^ located on a short α-helix that links the P+1 loop with the αF helix, Ser^557^ of the αI-helix, and 4 residues (Ser^602^, Thr^603^, Ser^604^, and Ser^612^) in the CT [21,59,65,130,132,133] (Figure 4). Among these phosphorylation sites, a total of 9 residues, including Ser^290^, Thr^312^, Thr^446^, Thr^449^, Thr^455^, and the 4 Ser/Thr residues in the CT, were discovered from immunoprecipitated BAK1 fusion proteins of transgenic Arabidopsis plants [21,134]. Whereas no single pTyr residue of BRI1 was detected by MS, a pTyr residue, Tyr^443^ located at the tip of the activation loop (Figure 4), was identified from MS analysis of an *E. coli* expressed recombinant BAK1 kinase [65]. Two additional Tyr residues, Tyr^463^ (located at a short α-helix that links the P+1 loop with the αF-helix) (Figure 4) and Tyr^610^ (on the CT that is not shown on the BAK1 kinase crystal structure), were previously shown to be phosphorylated based on immunoblotting analysis with anti-pTyr antibodies and loss-of-phosphorylation (Tyr-Phe) mutagenesis [73,135].

Structural analysis and molecular dynamics simulations suggested that the Thr^450^ phosphorylation is likely the most critical for BAK1 activation as its attached phosphate group likely binds to a positively-charged phosphate-binding pocket, which consists of Arg^415^ (within the HRD motif), Lys^439^ (on the β9-strand), and Arg^453^ (on the activation segment), to stabilize the activation segment (Figure 4). It is interesting to note that the third Arg residue of the BAK1’s phosphate binding pocket is from the activation segment itself instead the αC-helix that contributes the third Arg residue for the phosphate-binding pocket of the BRI1 kinase (Figure 2). The interaction between Arg^922^ with the phosphorylated Ser^1044^ is important to stabilize the N-lobe-C-lobe interaction in BRI1 to assemble the R-spine. It remains to be determined whether this is the likely cause for the apparent failure to incorporate a hydrophobic residue (likely Ile^338^) from the αC-helix to the R-spine of BAK1 and an apparent gap between the Lys^317^ (from the β3-strand) and Glu^334^ (of the αC-helix) that should form the conserved salt bridge important for the kinase activity. Consistent with the structural analysis, Thr^450^ was found to be essential for its autophosphorylation and transphosphorylation activities when assayed in vitro [59,132,136]. However, these results were contradictory to an earlier study demonstrating that the Thr^450^-Ala mutation only marginally affected the in vitro phosphorylation and in vivo signaling activities of BAK1 [21]. Further studies are needed to fully establish if the transphosphorylation of BAK1 by BRI1 at the Thr^450^ residue is absolutely required to activate the BAK1 kinase activity in vivo. It is also interesting to investigate if the gatekeeper residue Tyr^363^ might contribute to the assembly of the R-spine crucial for BAK1 activation (Figure 4).

Similar to what was discovered with the BRI1 kinase domain, the Thr residue in the P+1 loop, Thr^455^, is essential for the BAK1 activity. Mutating Thr^455^ to Ala or Asp/Glu resulted in a strong loss of BAK1’s phosphorylation activity in vitro and its BR-signaling function in transgenic Arabidopsis plants [21,59,132], suggesting that autophosphorylation of this highly conserved Ser/Thr residue might serve to inhibit rather activate BAK1 activity. Similar to what was discussed for the BRI1’s Thr^1049^ residue, the phosphorylation of Thr^455^ could interfere with ATP binding or substrate binding due to its strategic position at the P+1 loop. However, it remains a possibility that the phosphorylation of Thr^455^ is required for BAK1 activation but the Thr-Asp/Glu mutation might not be able to mimic the pThr^455^ residue at this strategic position. Site-directed mutagenesis coupled with in vitro phosphorylation assays and Arabidopsis transgenic experiments revealed another Ser/Thr residue whose phosphorylation serves to attenuate the BAK1 activity. Mutating Ser^286^ to Ala, which is located at the αB-helix capping the N-lobe, had little impact on the BAK1 activity; however, mutating Ser^286^ to Asp resulted in almost complete loss of in vitro phosphorylation activity of BAK1 and caused a dominant negative effect in transgenic Arabidopsis plants [21]. The impact of pSer^286^ on BAK1 activity is very similar to that of pThr^872^ on BRI1, which is also localized on the αB-helix that caps the BRI1’s N-lobe [22]. Further studies are needed to fully appreciate the negative impact of phosphorylating the Ser/Thr residues of the αB-helix. It is equally important to determine whether these Ser/Thr residues are intramolecularly autophosphorylated or intermolecularly transphosphorylated in order to fully understand the activation or autoinhibitory mechanisms of BAK1/SERKs and other LRR-RLKs.

In addition to the “*cis*” attenuation mechanism via auto/trans-phosphorylation of Ser^286^/Thr^455^, the BAK1 signaling activity could also be attenuated by a “*trans*” regulatory mechanism involving a phosphatase. A recent study suggested a role of a PP2A (consisting of subunits A1, C4, and B’η/ζ) in regulating the phosphorylation status of BAK1 in plant immunity response [137]. Given the demonstrated constitutive BAK1-PP2A association [137] and implication of PP2A in regulating BRI1 phosphorylation status [66], it is quite possible that this PP2A or its close homolog(s) could negatively influence BR signaling by controlling the phosphorylation level of the BRI1-associated BAK1/SERKs.

### 3.2. Regulating BAK1 Availability for BRI1 Interaction

Given the importance of BAK1/SERKs in activating the BR-bound BRI1 [49], it is no surprise that plant cells could regulate the availability of BAK1/SERKs to control BR signaling. The ligand-independent/dependent heterodimerization of BAK1 with BRI1 was previously known to induce endocytosis of both LRR-RLKs [15]. An Arabidopsis protein, MSBP1 (membrane steroid binding protein1) was previously found to inhibit BR signaling by stimulating BAK1 endocytosis to limit the amount of BAK1 on the PM [138]. Recently, several members of a small subfamily of LRR-RLKs known as BIRs were found to constitutively interact with BAK1/SERKs, thus interfering the signal initiation processes of many BAK1/SERKs-required LRR-RLKs [50,60,139,140,141,142,143]. However, only the BIR3-BAK1 interaction affects the BR-induced BRI1-BAK1/SERK heterodimerization and inhibits BR signaling [50]. A previously known gain-of-function mutant of BAK1, *elg1-D* (*elongated1-Dominant,* carrying an Asp^122^-Asn mutation) that was previously identified as a suppressor of an Arabidopsis gibberellin-deficient dwarf mutant [144,145], exhibits a much weaker binding affinity to the extracellular domain of BIR3 [143], thus increasing availability of BAK1 for BRI1 interaction to enhance BR signaling. However, it remains to be studied why overexpression of BIR3 led to a strong *bri1*-like dwarf phenotype with complete BR insensitivity but its loss-of-function *bir3* mutation enhanced the dwarf phenotype of a weak *bri1* mutant and exhibited no morphological similarity to the *elg1-D* mutant or weak phenotype of BRI1-overexpression. Further studies are also needed to fully understand why BIR3, but not its two other homologs, BIR1 and BIR2, can inhibit BR signaling. One possible explanation is its ability to interact with BRI1 and form ligand-independent BRI1-BAK1/SERKs-BIR3 receptor nanoclusters [60,95]. In addition to BIRs, members of another small subfamily of LRR-RLK [LRR-RLK IX subfamily, named BAK1-Associated Receptor-like Kinase1 (BARK1) and BARK1-Like Kinase 1-3 (BLK1-3)], could also bind BAK1/SERKs to influence BR signaling [146]. However, overexpression of BARK1 resulted in a hypersensitive phenotype, suggesting a potentially positive role of BARK1 in BR signaling. Unlike BIR3 that likely carries a cytoplasmic pseudokinase domain [50,147], the kinase domain of BARK1 is predicted to be an active kinase that could potentially help to phosphorylate and activate BAK1/SERKs or downstream signaling components to enhance BR signaling. A recent rice study also implicated OsREM4.1, a member of the remorin family thought to be associated with micro/nanodomains on the PM [97], as an interactor of OsSERK1 to interfere with the OsSERK1-OsBRI1 heterodimerization [96], suggesting that competitive BAK1/SERK-binding is likely a conserved mechanism to control BR signal initiation on the cell surface.

### 3.3. BAK1 Regulation by Other Mechanisms

Increasing evidence suggested an important role of S-glutathionylation, a post-translational modification of a Cys residue via its disulfide linkage with the Cys residue of glutathione (a γ-Glu-Cys-Gly tripeptide) [148], in regulating protein stability and activity in response to cellular oxidative stress. A recent in vitro study showed that BAK1 could be S-glutathionylated at Cys^353^ and Cys^408^ (shown in red spheres in Figure 4) by AtGRXC2 (*Arabidopsis thaliana* glutaredoxin C2) via a thiol-dependent reaction with glutathione disulfide, leading to a reduction of the in vitro BAK1’s kinase activity [149]. Molecular dynamics simulations suggested that S-glutathionylation of Cys^408^ promotes an inactive kinase conformation state [150]. The S-glutathionylation of Cys^408^ might directly affect the positioning of the αC-helix by steric hinderance or interfere with the interaction of the αC-β4 loop with the αE-helix, which is a crucial interaction that stabilizes the αC-helix to maintain an active kinase conformation [61]. It is interesting to note that Cys^408^ was mutated to Tyr in *bak1-5*, which compromises some plant innate immunity responses but has little effect on BR signaling [151]. It remains to be tested if BAK1/SERKs are S-glutathionylated in vivo and whether S-glutathionylation interferes their heterodimerization and transphosphorylation with BRI1 or other BAK1/SERKs-required LRR-RLKs and whether such a modification exhibits different impacts on BR signaling, plant immunity, and other BAK1/SERK-mediated processes.

A recent study suggested potential involvement of a Ca^2+^-dependent BAK1 proteolytic cleavage process in BR-mediated plant development and growth [152]. Biochemical studies suggested that the Asp^287^ residue (shown in green spheres in Figure 4), which is conserved among SERK family members and located right after the important regulatory Ser^286^ residue [17], is critical for its proteolytic cleavage. However, functional verification of the cleavage process in BR signaling was complicated by retention of a mutated BAK1 (carrying Asp^287^-Ala mutation) in the ER, most likely caused by ERQC-associated processes. This study suggested that considerable caution is needed in interpreting results with mutated transgenic constructs because some introduced mutations could potentially cause structural defects that could be detected by stringent quality control systems in plant cells, thus altering their protein abundance and/or subcellular locations.

One question that remains to be answered is whether or not BAK1/SERKs interact with downstream BR signaling components to directly influence the intracellular transmission of the extracellular BR signals in addition to their essential role of activating BRI1. Previous studies indicated that a C-terminally-tagged BAK1 or a *bak1* allele (known as *sobir7-1* for *suppressor of bir1 7-1,* carrying a non-sense mutation at Trp^597^ and lacking the CT) affected plant defense signaling but had little impact on BR signaling [131,153], leading to a conclusion that BAK1/SERKs require their CTs to interact with downstream signaling components to influence the plant immunity. It is interesting to note that the Arabidopsis BIK1 (Botrytis-Induced Kinase1), a receptor-like cytoplasmic kinase that is rapidly phosphorylated by FLS2-associated BAK1 and transduces the plant immunity signal to downstream targets [154], inhibits BR signaling by interacting with BRI1 but not the BRI1-interacting BAK1 [26]. Given the versatile roles of BAK1/SERKs in plant growth/development and plant defense [17], understanding the molecular mechanism(s) that determine the biochemical functions of BAK1/SERKs in BR signaling and FLS2/EFR-mediated plant defense responses will have a huge impact on plant biology.

## 4. BIN2, the Negative Regulator

The GSK3-like kinase BIN2 was originally discovered in a forward genetic screen for mutants similar to loss-of-function *bri1* mutants [30] and was subsequently shown to phosphorylate and negatively regulate BES1 and BZR1 [29,155,156] to block the intracellular transduction of the extracellular BR signals. The Arabidopsis genome encodes a total 10 GSK3-like kinases that are divided into four different subgroups [157], but genetic screens for BR-related dwarf mutants (*bin2* or *dwarf12*) [30,158] or leaf development mutants (*ucu* for *ultracurvata*) [159] discovered gain-of-function mutations only in BIN2 but not in any of the remaining 9 GSK3-like kinases. Interestingly, at least 7 of the 8 known *bin2*/*dwarf12*/*ucu1* mutants carry single AA changes in a 4-AA “TREE” (Thr^261^Arg^262^Glu^263^Glu^264^) motif that is a part of a surface-exposed α-helix (Figure 5) on the long connection fragment that links the αG and αH helices. However, subsequent reverse genetic studies showed that BIN2 functions redundantly with at least 6 other members of the Arabidopsis GSK3-like kinase family in regulating BR signaling [27,160,161,162,163].

### 4.1. BIN2 Regulation by Dephosphorylation

The signaling activity of BIN2 is likely regulated by post-translational modifications that include phosphorylation, acetylation, S-glutathionylation, S-nitrosylation, and ubiquitination. Like its animal homologs, BIN2 is a constitutively active kinase but is inactivated in response to BRI1/BAK1 activation. Molecular modeling revealed that BIN2 likely adopts an active kinase conformation with a well assembled R-spine consisting of Asp^223^ (the anchorage residue on the αF-helix), His^163^ (of the HRD motif), Phe^185^ (of the DFG motif), Met^85^ (of the αC-helix), and Leu^96^ (of the β4-strand), the Lys^69^-Glu^81^ salt bridge between the β3-strand and the αC-helix, and an opened activation segment (Figure 5). In animal cells, GSK3 phosphorylates its substrates via two different mechanisms: one requiring a priming phosphorylation of the substrate by a different kinase and the other requiring an adapter protein that binds both GSK3 and its substrate [164]. Accordingly, an animal GSK3 kinase can be inactivated by two general mechanisms: phosphorylation of a key Ser/Thr residue in its autoinhibitory N-terminal fragment and interaction of a GSK3-binding protein [164]. The phosphorylated N-terminal fragment competes with a “primed” substrate for the phosphate-binding pocket consisting of three positively-charged residues, Arg^96^, Arg^180^, and Lys^205^ in the human GSK3β (corresponding to Arg^80^, Arg^164^, and Lys^189^ in BIN2, respectively) (Figure 5), while a GSK3-binding protein competitively prevents the GSK3-substrate binding necessary to phosphorylate a non-primed GSK3 substrate.

There has been no evidence so far for a similar phosphorylation-mediated autoinhibitory mechanism in plant cells to inactivate BIN2 in response to BR. An earlier biochemical study showed that the BIN2-catalyzed BES1/BZR1 phosphorylation does not need a priming phosphorylation but instead requires a direct binding between BIN2 and BES1/BZR1 that carry a 12-AA docking motif [165]. However, phosphorylation is involved in BIN2 regulation, which is likely mediated by several protein Ser/Thr phosphatases (PSPs) instead of protein kinases [27,166]. It is widely believed that BIN2 is inhibited through dephosphorylating a phosphorylated Tyr residue (pTyr^200^) in the activation segment by BSU1/BSLs [27]. The phosphorylation of the corresponding Tyr residue in the human GSK3 kinases was known to occur intramolecularly during HSP90-facilitated folding process [167,168], and peptides of BIN2 or its closest homologs containing the pTyr^200^ residue were identified previously by phosphoproteomic studies [169,170]. However, the requirement of the pTyr residue (pTyr^216^ in human GSK3β) for the GSK3 kinase activity remains controversial for animal GSK3 kinases [171]. An early study reported that the mammalian GSK3 kinases required the pTyr residue for their maximum enzymatic activities [172], but later studies showed that non-phosphorylated GSK3β adopted an active conformation [173] and mutating the Tyr residue to Phe had a marginal impact on the GSK3β kinase activity [174,175]. As expected from published GSK3 structures, structure modeling of BIN2 suggests that Tyr^200^ adopts an *anti*-conformation (away from the substrate binding site) to avoid steric clash with a substrate (Figure 5). However, mutating Tyr^200^ to Phe was shown to completely inactivate BIN2 in transgenic Arabidopsis plants [27], but the inhibitory impact of the Tyr-Phe mutation could be caused by lacking of a hydroxyl group or a phosphate group on the aromatic ring as the Tyr residue is absolutely conserved among GSK3 and GSK3-like kinases [158]. Thus, further investigation is needed to confirm whether dephosphorylation of the pTyr^200^ residue is capable of complete inactivating BIN2.

The current BR signaling model suggests that the phosphate group of pTyr^200^ is removed by BSU1/BSLs [27], members of a plant-specific PPKL family [176]. BSU1, which was originally discovered as an activation-tagged *bri1* suppressor [177] but later found to be exclusively present in the Brassicaceae family and exclusively expressed in pollen with yet unknown physiological function [178], was predicted and shown to be a functional PSP [176,177]. Consistently, molecular modeling revealed that BSU1 has the conserved PSP structure of an α/β fold with a β-sandwich surrounded by a large and a small α-helical domains (Figure 6) and contains three catalytic signature motifs, GDXHG (GlyAsp^510^IleHis^511^Gly in BSU1), GDXVDRG (GlyAsp^544^TyrValAspArgGly in BSU1), and GNHE (GlyAsn^576^HisGlu in BSU1), plus two conserved His residues (His^629^ and His^707^ in BSU1). It was well established that these conserved residues coordinate two metal ions that bind and activate a water molecule for its nucleophilic attack on the phosphate ester linkage with a Ser/Thr residue [179]. These signature motifs are quite different from the HCX_5_R motif (Figure 6), the catalytic signature of PTP/DSPP with the Cys residue being the enzymatic nucleophile and Arg directly binding the phosphate group of pTyr/pSer/pThr [180]. Furthermore, a previous genetic study coupled with phylogeny/evolution analysis of BSU1/BSLs questioned a major BR signaling role of the pollen-expressed BSU1 and its three homologs whose loss-of-function mutations had a marginal impact on the in vivo BIN2 activity [178]. It is important to note that a role of BSU1/BSLs in BR signaling was supported by a recent study in rice that investigated genetic and biochemical interactions between a rice homolog of BSL1, qGL3 (quantitative trait locus regulating grain length3, also known as OsPPKL1), and OsGSK3 (a rice homolog of BIN2) [181]. However, the rice study did not test whether qGL3 was capable of dephosphorylating the conserved Tyr residue of OsGSK3. Given the crucial role of BIN2 and its homologs in regulating the intracellular transduction of the extracellular BR signals into the nucleus, further genetic, biochemistry, and structural studies are needed to fully understand the biochemical roles of pTyr^200^ and BSU1/BSLs in BR signaling and other relevant physiological processes. The results of these future studies will not only increase our knowledge of BR signaling, but will also significantly enhance our understating of GSK3 regulation in plants and catalytic mechanism of the plant specific PPKLs.

BIN2 could also be regulated through dephosphorylation by ABI1 (ABA-Insensitive1) and ABI2 [166], two protein phosphatase 2C-type PSPs known to play inhibitory roles in transducing the signal of the plant stress hormone abscisic acid (ABA) [182]. While this discovery seems to support earlier findings that the ABA-BR antagonism is mediated by a biochemical event located between BRI1 activation and BIN2 inhibition of the BR signaling cascade [183], it remains to be determined which phosphorylated residue(s) are dephosphorylated by ABA1/2. Quantitative analysis of the in vivo BIN2 phosphorylation dynamics in response to ABA and/or BR treatment coupled with structural analysis of BIN2 will not only lead to a better understanding of the well-known ABA-BR antagonism but will also provide biochemical insights into the regulatory mechanism(s) of plant GSK3-like kinases.

### 4.2. BIN2 Regulation by Other Post-Translational Modifications

In additional to phosphorylation/dephosphorylation, BIN2 can be regulated by other post-translational modifications. A recent study implicated a role of an Arabidopsis histone deacetylase HDA6 (histone deacetylase6) in BIN2 inactivation [184]. HDA6 interacts with BIN2 and deacetylates BIN2 at Lys^189^ in vitro. Because Lys^189^ is one of the positively charged residues that make up the highly conserved phosphate-binding pocket (Figure 5), its acetylation should neutralize its positive charge and likely reduce the BIN2 activity whereas deacetylation of the acetylated Lys^189^ is expected to increase the BIN2 activity via enhanced BIN2 binding with BES1/BZR1 that has multiple conserved (Ser/Thr)X_3_(Ser/Thr) GSK3 phosphorylation repeats. Consistent with the published GSK3 structures and the role of the conserved phosphate binding pocket [173,185], acetylation of Lys^205^ in the human GSK3β (corresponding to Lys^189^ of BIN2) was found to reduce its phosphorylation activity while a histone deacetylase sirtuin1 (Sirt1) and the Lys^205^-Arg mutation increased the GSK3β activity [186]. By contrast, the Lys^189^-Arg mutation, which should eliminate its presumed acetylation, actually reduced the in vitro kinase activity and in vivo signaling function of BIN2 [184], suggesting that acetylation of Lys^189^ stimulates BIN2 activity via a yet to be defined biochemical mechanism in plants. However, the mutagenesis results should be interpreted with caution as previous sequence analyses showed that Lys^189^ is absolutely conserved between BIN2 and its homologs from other plants or other eukaryotic organisms [30,158]. By contrast, both Arg and Lys were found at the corresponding position in plant LRR-RLKs [9,187]. It is noteworthy that acetylation at another conserved Lys residue, Lys^183^ (corresponding to Lys^167^ in BIN2) known to be important for ATP binding was recently shown to inhibit the mammalian GSK3β activity, whereas its deacetylation by another histone deacetylase Sirt7 served to promote ATP binding and to stimulate GSK3 activity [188]. Further studies, such as mapping the in vivo BIN2 acetylation sites and identification of BIN2 acetylation enzyme(s), are needed to fully understand the role of acetylation and deacetylation in BIN2 regulation.

Nitric oxide (NO), an important signaling molecular, was shown to inhibit the kinase activity of BIN2 by S-nitrosylation at the conserved Cys^162^ site in an in vitro assay [189]. It is important to note that Cys^162^ sits right before the conserved HRD motif (Figure 5) and its S-nitrosylation could thus interfere with the assembly of the R-spine (involving the His^163^ residue) or the interaction between a phosphate group of BES1/BZR1 with the Arg^164^ residue, one of the three positively charged residues of the BIN2’s phosphate binding pocket (Figure 5). S-nitrosylation was recently shown to regulate the human GSK3β activity in a very interesting way that inhibits its phosphorylation of cytosolic substrates containing the (Ser/Thr)X_3_(Ser/Thr) phosphorylation motif but promotes its nuclear localization to phosphorylate its nuclear substrates containing the (Ser/Thr)-Pro motif [190]. In addition to the NO-triggered S-nitrosylation of Cys residues, a recent study suggested that certain cysteine residues of BIN2, such as Cys^59^ (sits at the end of the β2-strand) and Cys^162^, could also be S-glutathionylated in vitro and are likely involved in the reactive oxygen species (ROS)-induced activation of BIN2 in vivo [191]. Further studies are needed to confirm these S-nitrosylation and S-glutathionylation sites in vivo and to investigate their biochemical/cellular impacts on the catalytic activity and subcellular localization of BIN2 and other plant GSK3-like kinases. It will also be interesting to determine if additional post-translational modifications, such as ribosylation, SUMOylation, and methylation that were implicated in regulating the mammalian GSK3 kinases [171], are also involved in regulating the in vivo BIN2 activity.

### 4.3. BIN2 Regulation by Protein–Protein Interactions

Given the earlier discovery of a direct BIN2-BES1/BZR1 binding [165] that was recently confirmed by a single-molecular analysis [191], it is no surprise that BIN2 is also regulated by its competitive binding with other cellular proteins in plant cells. Recent studies showed that BIN2 interacted in vivo with several well-studied light signaling components, including CRY1 (Cryptochrome1, a blue-light photoreceptor) [192], the COP1 (Constitutive Photomorphogenesis1)/SPA (Suppressor of *phyA-105*) complex (a light regulated E3 ubiquitin ligase) [193], and HY5 (long hypocotyl5, a bZIP-type transcriptional factor) [194]. CRY1 exhibits a blue light-dependent binding with both BIN2 and BZR1 to enhance the BIN2-catalyzed BZR1 phosphorylation [192], thus inhibiting its nuclear translocation (likely by increased interaction with 14-3-3 proteins [195]) and its DNA binding activity through CRY1’s competitive binding to the BZR1’s DNA binding domain. Thus, the blue light-activated CRY1 functions as an adapter to further promote the recruitment of BZR1, which carries a BIN2-binding motif near its C-terminus [165], to the BIN2 kinase. While the blue light-dependent CRY1-BIN2 interaction stimulates the BIN2-catalyzed BZR1 phosphorylation, COP1/SPA was recently found to inhibit the BIN2-catalyzed phosphorylation of PIF3 (phytochrome interacting factor3) [193]. COP1/SPA1 is a well-studied crucial photomorphogenesis repressor complex [196] that was previously shown to interact with CRY1 [197,198], while PIF3 functions together with its homologs to repress photomorphogenesis in the dark [199]. Despite being an E3 ubiquitin ligase [196], COP1/SPA did not promote BIN2 degradation to inhibit its PIF3 phosphorylation activity. Instead, the inhibitory effect of the COP1/SPA complex on the BIN2-catalyzed PIF3 phosphorylation is mediated by preventing the BIN2-PIF3 binding via competitive bindings of COP1/SPA with the kinase and the substrate [193]. Further experiments are needed to see whether these two studies actually revealed two general mechanisms to regulate the phosphorylation activities of BIN2 and other plant GSK3-like kinases: one using an adapter protein that facilitates the BIN2-substrate phosphorylation while the other relying on a disrupter that interferes the BIN2-substrate binding.

A more interesting discovery on the impact of protein–protein interaction on the BIN2 phosphorylation activity came from a recent study [194] that investigated the genetic and biochemical interactions between BIN2 and HY5, a bZIP-type transcription factor with versatile roles in plant growth and development [200] and a known substrate of the COP1/SPA E3 ligase [201]. Biochemical experiments coupled with molecular modeling and computational simulations suggested that the HY5-BIN2 binding stimulated the BIN2 catalytic activity assayed by a HY5-stimulated increase in pTyr^200^ signal [194], suggesting that HY5 might function similarly as the mammalian HSP90 known to facilitate GSK3β folding during which its Tyr^206^ (the equivalent of BIN2’s Tyr^200^ residue) was intramolecularly autophosphorylated [167]. Further experimentation is needed to confirm its hypothetic “chaperone-like” function to assist BIN2 folding into an active kinase in vivo.

### 4.4. BIN2 Regulation by Subcellular Localization

Given that many of the known BIN2 substrates are transcriptional factors [157], it is no surprise that differential subcellular localization is an important regulatory mechanism to control the BIN2 phosphorylation activity in vivo. An earlier study with the GFP-tagged BIN2 or the mutant bin2-1 [carrying the Glu^264^-Lys mutation in the TREE motif (Figure 5)] revealed that the bin2-1 protein mainly accumulated in the nucleus while the wild-type BIN2 seemed to exhibit more or less equal distributions at the PM, in the cytosol and the nucleus [163]. More importantly, adding a nuclear localization signal (NLS) to the wild-type BIN2 resulted in a stronger BIN2 activity to block BR signaling, whereas fusing a nuclear export signal (NES) greatly reduced its BR signaling-inhibition activity. Consistent with these findings, Arabidopsis OPS (OCTOPUS), a PM-associated protein crucial for phloem differentiation in Arabidopsis roots [202], binds and recruits BIN2 to the PM, thus reducing the BIN2 phosphorylation activity towards BES1/BZR1 and enhancing BR signaling to promote phloem differentiation [203]. Similarly, a stomata lineage cell-expressed POLAR (polar localization during asymmetric division and redistribution) protein [204], interacts with BIN2 and several other GSK3-like kinases to prevent their nuclear localization and transiently polarizes their subcellular distributions, thus driving asymmetric cell division essential for stomata development [205]. It is noteworthy that the rice PM-associated protein GW5 (grain width and weight5) interacts with OsGSK2, a rice homolog of BIN2, to inhibit the OsGSK2’s phosphorylation activity, presumably by recruiting OsGSK2 to the PM, and to enhance BR signaling that regulates grain width and weight in rice [206]. The regulation of BIN2/GSKs via differential subcellular localization was also discovered in *Sorghum bicolor*, which uses DW1 (Dwarfing1), a PM/cytosol-localized protein, to interact with SbBIN2 (a *Sorghum bicolor* BIN2 homolog), thus preventing the nuclear localization of SbBIN2 and increasing BR signaling [207]. It will be interesting to know if their Arabidopsis homologs can also bind BIN2/GSK3s to enhance BR signaling. More importantly, further studies are needed to understand how these four distinct proteins with little sequence homology interact with BIN2/GSK3s to regulate their subcellular distributions. Careful biochemical analyses of their association with BIN2/GSK3s could define the minimum binding motifs while structural biology studies might reveal similar docking mechanisms to allow their BIN2/GSK3-binding.

In contrast to these four proteins that enhance BR signaling by preventing nuclear localization of BIN2/GSK3s, at least two members of the Arabidopsis HSP90 family whose mammalian homologs were known to assist the folding and activation of GSK3β [167], were shown to keep BIN2 inside the nucleus in an ATP-dependent manner to inhibit BR signaling [208]. Treatment with geldanamycin, a widely used inhibitor of HSP90-dependent ATPase, or active BR, resulted in nuclear export of the HSP90-BIN2 complexes [208]. Further studies are needed to determine 1) whether or not the Arabidopsis HSP90s are needed for the cytosolic folding and autoactivation of BIN2/GSK3s, 2) if the HSP90-bound BIN2 is still capable of phosphorylating its many substrates, and 3) how the extracellular BR signal is rapidly sensed by the nuclear-localized HSP90-BIN2 complexes that translocate into the cytosol [208].

### 4.5. BIN2 Inhibition by Degradation

Proteasome-mediated BIN2 degradation is another important mechanism to keep this negative regulator of BR signaling at low levels. An earlier study suggested that the *bin2-1* (Glu^264^-Lys) mutation greatly stabilized the mutant kinase, which was partially responsible for its increased BES1/BZR1 phosphorylation activity [209]. A recent study demonstrated that the Arabidopsis KIB1, a Kelch-repeat-containing F-box E3 ubiquitin ligase that was identified as a genetic suppressor of a constitutively active form of BZR1, bzr1-1D, ubiquitinated BIN2 in vitro and stimulated the proteasome-mediated BIN2 degradation in vivo [28]. In addition to its role in BIN2 degradation, the Kelch repeat-mediated KIB1-BIN2 interaction blocked the BIN2-BZR1 binding, thus further reducing the BIN2 phosphorylation activity [28]. It remains to be determined whether proteasome-mediated degradation is an evolutionally conserved mechanism in other plant species to regulate the phosphorylation activity of BIN2 homologs given a recent discovery of restricted distribution of KIB1 and its homologs in the Brassicaceae family and their extremely low expression in vegetative tissues [210]. It is interesting to note that the proteasome-mediated GSK3 degradation was previously shown to be induced by glucocorticoid in mammalian cells although the identity of a GSK3-interacting E3 ubiquitin ligase remains unknown [211].

## 5. Conclusions and Remarks

The basic scheme of the BR signaling pathway is quite simple, involving a PM-localized BR receptor complex, a cytosolic/nuclear-localized inhibitor BIN2, and two key transcription factors, BES1 and BZR1. In the absence of BR, BIN2 is constitutively active to phosphorylate and inhibit BES1/BZR1. BR binding to BRI1 triggers its conformational changes to allow stable heterodimerization, transphosphorylation, and activation of BRI1 and BAK1, leading to inhibition of BIN2 and nuclear accumulation of non/de-phosphorylated BES1/BZR1. These BES1/BZR1 proteins bind to their target promoters to control expression of thousands of genes important for plant growth, development, and stress tolerance. Studies in the last twenty years have provided molecular understanding of key BR signaling events and discovered a wide range of biochemical and cellular mechanisms that regulate the abundance, subcellular locations, and biochemical activities of these key signaling components, which have dramatically enhanced our understanding of BR signaling processes. Despite these great achievements, there are still outstanding questions and unresolved controversies (discussed throughout this review) that require further investigation. Emerging technologies and novel experimental approaches will not only help to answer questions and resolve controversies, but will also lead to new discoveries that will provide a high-resolution atomic portrait of the BR signaling events to further our understanding of the BR signaling pathway and its interactions with other plant signaling processes.

## Figures and Tables

**Figure 1 ijms-21-04340-f001:**
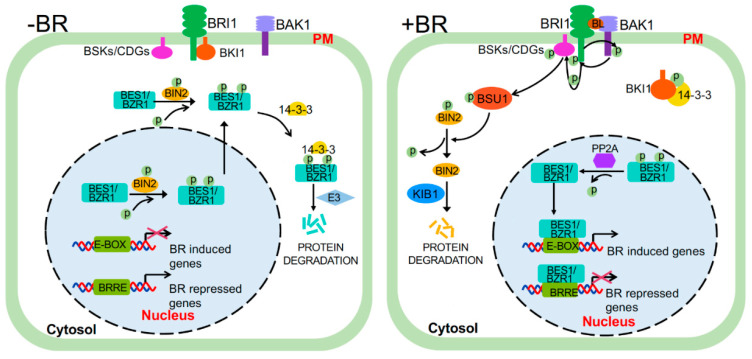
A current model of BR signaling. When BRs (brassinosteroids) are absent (left), BRI1 (Brassinosteroid-Insensitive1) is kept inactive by its autoinhibitory C-terminus and BKI1 (BRI1 Kinase Inhibitor1) association. BIN2 (Brassinosteroid-Insensitive2) is constitutively active and phosphorylates BES1 (bri1-EMS suppressor1)/BZR1 (Brassinazole-resistant1) transcription factors to promote their degradation and 14-3-3-mediated cytosolic retention, and to directly inhibit their DNA-binding activities. When BRs are present (right) and bind to the extracellular domains of BRI1 and its co-receptor BAK1 (BRI1 Associated receptor Kinase1) to activate the two receptor kinases, leading to dissociation of BKI1 from BRI1, phosphorylation and activation of BSKs (BR-signaling kinases)/CDGs (Constitutive Differential Growth) and BSU1 (bri1 suppressor1). The activated BSU1 dephosphorylates and inactivates BIN2 while KIB1 (Kink suppressed in *bzr1-1D*1) promotes BIN2 degradation, causing nuclear accumulation of PP2A (protein phosphatase 2A)-dephosphorylated BES1/BZR1 that bind BRRE (BR response element)/E-box-containing promoters to regulate expression of thousands of BR-responsive genes important for plant growth and development.

**Figure 2 ijms-21-04340-f002:**
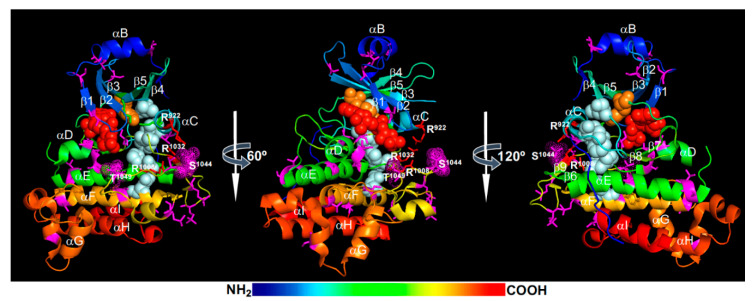
A structural model of the activated BRI1 kinase domain. Shown here are three (0°, 60°, and 180°) rotational views of a rainbow-colored ribbon model of the crystal structure of the BRI1 kinase domain (Protein Data Bank No. 4oh4). Individual α-helices (αB-αI) and β-strands (β1-β5) were labelled. The magenta-colored sticks indicate phosphorylated Ser/Thr residues with the magenta dots surrounding Ser^1044^ and Thr^1049^ of the activation segment, the green sticks denote the Lys^911^ and Glu^927^ residues that form the salt bridge between the β3-strand and αC-helix, the red sticks mark the three positively charged residues of the phosphate-binding pocket, and the orange sticks represent the phosphorylated Tyr residue. The red spheres show adenylyl-imidodiphosphate (a non-hydrolysable ATP analog), the orange spheres indicate the gatekeeper Tyr^956^ residue that is also phosphorylated in in vitro assays, and the light-blue spheres denote the five regulatory-spine (R-spine) residues (from lower to upper: Asp^1068^, His^1077^, Phe^1028^, Ile^931^, and Leu^942^). The rainbow bar indicates the order of amino acids (AAs) from the N-terminus (blue) to the C-terminus (red).

**Figure 3 ijms-21-04340-f003:**
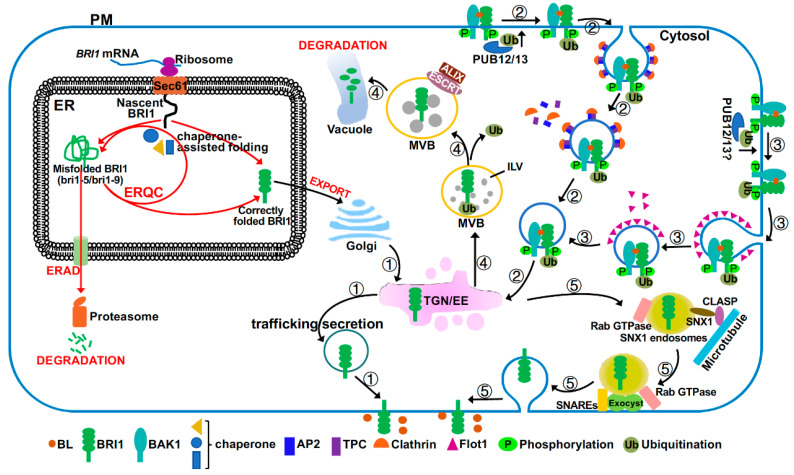
A current model of regulating the BRI1 abundance on the plasma membrane (PM). Newly synthesized BRI1 is translocated through the Sec61 translocon from the ER (endoplasmic reticulum)-associated ribosomes into the ER where it undergoes chaperone-assisted protein folding. Correctly folded BRI1 is exported into the Golgi to continue its secretory journey through the TGN (*trans*-Golgi network)/EE (early endosome) to be targeted on the PM (1), whereas the incorrectly/misfolded BRI1, such as its mutant variants bri1-5 and bri1-9, are retained in the ER by ERQC (ER quality control) for refolding and degradation via ERAD (ER-associated degradation) that involves retrotranslocation and cytosolic proteasome. The PM-localized BRI1 or BRI1/BAK1 heterodimer is presumably ubiquitinated by the PUB12/13 (plant U-box protein12/13) E3 ligases and undergoes constitutive CME (clathrin-mediated endocytosis)-mediated internalization (2) or ligand-induced CIE (clathrin-independent endocytosis)-mediated internalization (3). The endocytosed BRI1 at the TGN/EE could be packaged into ILVs (intraluminal vesicles) and delivered to the vacuole for degradation via ESCRT (endosomal sorting complex required for transport)-mediated biogenesis of MVBs (multivesicular bodies) and eventual MVB-vacuole fusion (4). Alternatively, the TGN/EE-localized BRI1 can be recycled back to the PM via retromer-mediated cargo selection, microtubule-assisted vesicle trafficking, and exocyst/SNARE (soluble N-ethylmaleimide-sensitive factor-attachment protein receptor)-involved exocytosis (5). The circled numbers indicate different secretion/trafficking routes. SNX1 (sorting nexin1) is one of the core retromer subunits, CLASP (cytoplasmic linker-associated protein) is a microtubule-associated protein, and ALIX (the Arabidopsis homolog of apoptosis-linked gene 2-interacting protein X) is a cytosolic ESCRT-associated protein.

**Figure 4 ijms-21-04340-f004:**
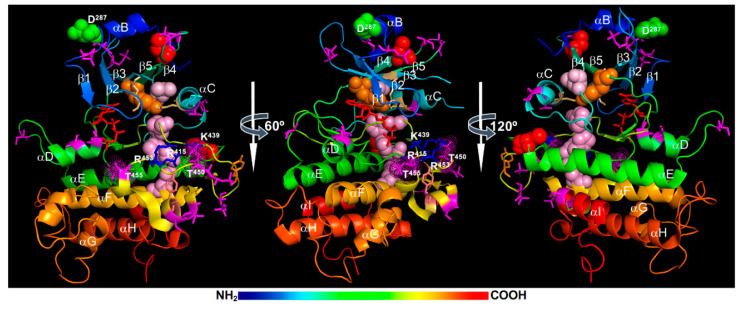
A crystal structure of the BAK1 kinase domain. Shown here are three rotation (0°, 60°, and 180°) views of a rainbow-colored ribbon model of the crystal structure of the BAK1 kinase domain (Protein Data Bank No: 3uim). Individual α-helices (αB-αI) and β-strands (β1-β5) were labelled. The purple sticks indicate phosphorylated Ser/Thr residues with the Thr^450^ and Thr^455^ residues surrounded with purple dots. The red sticks show the bound adenylyl-imidodiphosphate (a non-hydrolysable ATP analog), the light-orange sticks denote the two residues that form the conserved salt bridge between the β3-strand and the αC-helix, the blue sticks represent the three positively charged residues that make up the phosphate-binding pocket, and the dark-orange sticks indicate phosphorylated Tyr residues. The pink spheres mark the R-spine residues, the red spheres show the two Cys residues that were S-glutathionylated in vitro, the orange spheres denote the gatekeeper Tyr residue, and the green spheres designate the Asp^287^ residue important for Ca^2+^-dependent BAK1 cleavage. The rainbow bar indicates the order of AAs from the N-terminus (blue) to the C-terminus (red).

**Figure 5 ijms-21-04340-f005:**
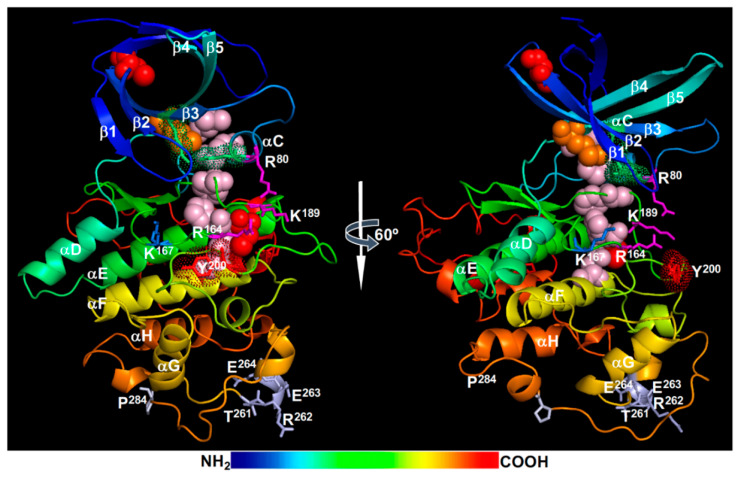
A structural model of BIN2. Shown here are two rotational (0° and 60°) views of a modeled BIN2 structure obtained at SWISS-MODEL (https://swissmodel.expasy.org/). Individual β-strands (β1-β5) and α-helices (αC-αH) were labeled. The two extra antiparallel-β-strands that cap the N-lobe are not labelled. The TREE motif and the Pro^284^ residue mutated in known *bin2*/*dwarf12*/*ucu1* alleles are indicated with grey-colored sticks. The magenta sticks show the three positively charged residues [Arg^80^, Arg^164^, and Lys^189^ (also acetylated)] of the conserved phosphate-binding pocket, the red sticks with red dots denote the Tyr^200^ residue, the green sticks with green dots mark the Lys^69^-Glu^81^ salt bridge between the β3-strand and the αC-helix, and the blue sticks represent the Lys^167^ residue that corresponds to the acetylated Lys^183^ residue of the human GSK3β. The red spheres show the Cys residues that were shown to be S-nitrosylated or S-glutathionylated in vitro, the orange spheres indicate the gatekeeper Met^115^ residue, and the pink spheres mark the 5 R-spine residues (from lower to top: Asp^223^, His^163^, Phe^185^, Met^85^, and Leu^96^). Important residues are also labelled with single letter codes with positions. The rainbow bar indicates the order of AAs from the N-terminus (blue) to the C-terminus (red).

**Figure 6 ijms-21-04340-f006:**
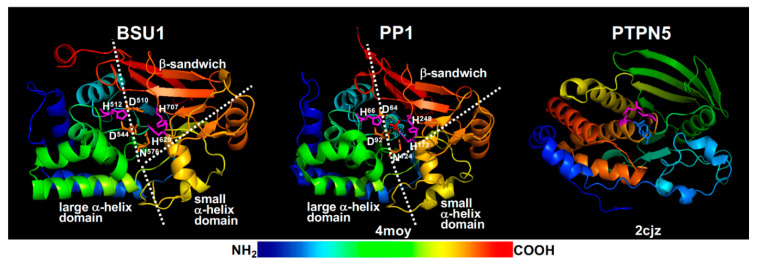
Comparison between a structure model of BSU1’s phosphatase domain with crystal structures of PSP (protein Ser/Thr phosphatase) and PTP (protein tyrosine phosphatase). Shown here are a predicted structure of the BSU1’s C-terminal phosphatase domain [obtained at SWISS-MODEL at https://swissmodel.expasy.org/ using the protein sequence of BSU1 (accession No: NP_171844)], a crystal structure of a human PSP (HsPP1; Protein Data Bank No. 4moy), and a crystal structure of a human PTP (HsPTPN5 for the human PTP non-receptor type 5; Protein Data Bank No. 2cjz). The conserved metal-coordinating amino acid residues are indicated by colored sticks: orange colored Asp(D)^510^, Asp(D)^544^, and Asn(N)^576^, and three magenta-colored His(H) residues. The three dotted white lines separate the β-sandwich of two β-sheets from the two flanking α-helix domains in the structure models of BSU1 and PP1. The dotted spheres and the red sticks in PP1 represent two metal ions and the phosphate, respectively. The two conserved residues (Cys and Arg of the HCX_5_R catalytic signature motif) are colored with magenta in the human PTPN5 structure and a phosphorylated tyrosine is shown with blue sticks. The rainbow bar indicates the order of AAs from the N-terminus (blue) to the C-terminus (red) of each peptide.

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
