# Peer review of "Regulation of Three Key Kinases of Brassinosteroid Signaling Pathway"

_ijms, 2020, doi:10.3390/ijms21124340_

Round 1
Reviewer 1 Report
The reviewed article presents in detail all the transduction motifs of brassinosteroid signal on the example of Arabidopsis thaliana. A critical approach to the molecular model of these hormones makes the article extremely interesting for every reader. In a clear and straightforward way it describes the connection of brassinosteroids with receptors and transmission of this signal to subsequent elements. The figures presented are within the scope and well-prepared, especially Figure 3, because rarely is a model of regulating the BRI1 abundance on the plasma membrane shown. I believe that the reviewed article has no weaknesses; of course, I found some editorial shortcomings, but they can be corrected in the author's proof. Generalizing, the manuscript is well organized and written. I would like to recommend to publish this manuscript in the present state.
Author Response
The reviewer #1 recommended that “extensive editing of English language and style required”. To address this recommendation, I worked closely with Prof. Erik Nielsen, a Native English-speaking colleague of mine at University of Michigan, corrected many grammatical errors, and made many style changes throughout the manuscript. A pdf file with highlighted changes is submitted.
Reviewer 2 Report
The presented manuscript is interesting and should be accepted for printing after minor changes.
Here are aspects to consider:
Title: I suggest changing the title to something like this: "Regulation of the plant kinase activity of brassinosteroid signaling pathway".
Figure 1 and 3: Please explain here all the abbreviations used in the Figures 1 and 3 so that these figures are self-explanatory.
Author Response
- This reviewer suggested that we change the title to “Regulation of the plant kinase activity of brassinosteroid signaling pathway”. Because our manuscript also discussed regulation of protein abundance and subcellular localization, we decided to change the title from “Regulation of three kinases of brassinosteroid signaling” to “Regulation of three key kinases of brassinosteroid signaling pathway”.
- We also provided the full names of all the abbreviations used in the legends of Figure 1 and 3 as suggested by this reviewer.